# Obesity during Pregnancy in the Horse: Effect on Term Placental Structure and Gene Expression, as Well as Colostrum and Milk Fatty Acid Concentration

**DOI:** 10.3390/vetsci10120691

**Published:** 2023-12-04

**Authors:** Morgane Robles, Delphine Rousseau-Ralliard, Cédric Dubois, Tiphanie Josse, Émilie Nouveau, Michele Dahirel, Laurence Wimel, Anne Couturier-Tarrade, Pascale Chavatte-Palmer

**Affiliations:** 1BREED, Domaine de Vilvert, Université Paris Saclay, UVSQ, INRAE, 78350 Jouy en Josas, France; delphine.rousseau@inrae.fr (D.R.-R.); anne.couturier-tarrade@inrae.fr (A.C.-T.); 2BREED, Ecole Nationale Vétérinaire d’Alfort, 94700 Maisons-Alfort, France; 3Institut Polytechnique Unilasalle, 76130 Mont-Saint-Aignan, France; 4Institut Français du Cheval et de l’Equitation, Station Expérimentale de la Valade, 19370 Chamberet, Francelaurence.wimel@ifce.fr (L.W.)

**Keywords:** horse, obesity, milk, placenta, fatty acid, stereology, colostrum

## Abstract

**Simple Summary:**

The prevalence of obesity in horses is high, but the impact of obesity during pregnancy on placental function and milk production is currently unknown in this species. This paper is a follow-up study of previous work that showed that maternal obesity during gestation affected the glucose metabolism, systemic inflammation, and osteoarticular lesions in growing foals. At birth, placentas were collected for histology and qPCR analysis, and until 3 months of lactation, the colostrum, milk, and plasma of foals and mares were sampled for fatty acid profile analyses. No effects of obesity were observed for placental analyses. During the 2nd and 3rd months of lactation, mares and foals suffered heat stress as a strong heat wave, followed by a drought, occurred. According to the results previously observed in the foals, colostrum as well as the milk of obese mares had a more pro-inflammatory profile and indicated a decreased ability to adapt to heat stress in this group.

**Abstract:**

In horses, the prevalence of obesity is high and associated with serious metabolic pathologies. Being a broodmare has been identified as a risk factor for obesity. In other species, maternal obesity is known to affect the development of the offspring. This article is a follow-up study of previous work showing that Obese mares (O, n = 10, body condition score > 4.25 at insemination) were more insulin resistant and presented increased systemic inflammation during pregnancy compared to Normal mares (N, n = 14, body condition score < 4 at insemination). Foals born to O mares were more insulin-resistant, presented increased systemic inflammation, and were more affected by osteoarticular lesions. The objective of the present study was to investigate the effect of maternal obesity on placental structure and function, as well as the fatty acid profile in the plasma of mares and foals, colostrum, and milk until 90 days of lactation, which, to our knowledge, has been poorly studied in the horse. Mares from both groups were fed the same diet during pregnancy and lactation. During lactation, mares were housed in pasture. A strong heat wave, followed by a drought, occurred during their 2nd and 3rd months of lactation (summer of 2016 in the Limousin region, France). In the present article, term placental morphometry, structure (stereology), and gene expression (RT-qPCR, genes involved in nutrient transport, growth, and development, as well as vascularization) were studied. Plasma of mares and their foals, as well as colostrum and milk, were sampled at birth, 30 days, and 90 days of lactation. The fatty acid composition of these samples was measured using gas chromatography. No differences between the N and O groups were observed for term placental morphometry, structure, or gene expression. No difference in plasma fatty acid composition was observed between groups in mares. The plasma fatty acid profile of O foals was more pro-inflammatory and indicated an altered placental lipid metabolism between birth and 90 days of age. These results are in line with the increased systemic inflammation and altered glucose metabolism observed until 18 months of age in this group. The colostrum fatty acid profile of O mares was more pro-inflammatory and indicated an increased transfer and/or desaturation of long-chain fatty acids. Moreover, O foals received a colostrum poorer in medium-chain saturated fatty acid, a source of immediate energy for the newborn that can also play a role in immunity and gut microbiota development. Differences in milk fatty acid composition indicated a decreased ability to adapt to heat stress in O mares, which could have further affected the metabolic development of their foals. In conclusion, maternal obesity affected the fatty acid composition of milk, thus also influencing the foal’s plasma fatty acid composition and likely participating in the developmental programming observed in growing foals.

## 1. Introduction

Alterations in the early environment, from conception until the end of puberty, have been demonstrated to affect the development of the individual and his health in adulthood in both humans and animal models. The developmental origins of health and diseases (DOHaD) can also be applied to the horse [1,2,3,4], as embryo transfer [5,6,7,8,9], maternal overnutrition [10,11,12,13], undernutrition [14,15,16], and obesity [17] have been shown to affect the development of foals until at least 2 years of age. 

Obesity is becoming increasingly prevalent in the equine industry. Studies conducted worldwide have reported that the prevalence of overweight horses ranges from 2% to 72%, while obesity is observed in 1% to 19% of horses. The variation in prevalence depends on factors such as the country, the defined threshold for a body condition considered overweight or obese, the season, and the intended use of the horses [18,19]. Research has shown that horse owners often have poor knowledge of the nutritional needs of their horses [20] and systematically underestimate their horses’ body condition. Furthermore, horse owners tend to believe that weight management can have immediate negative effects on a horse’s well-being [21]. These misconceptions can contribute, along with factors like inactivity and genetic predispositions, to the high prevalence of obesity observed in the equine population. 

Obesity in horses has been associated with metabolic alterations and diseases, including insulin resistance [22], alterations of adipose tissue and cardiovascular function [23,24], low-grade inflammation [25] and weakened immunity [26], equine metabolic syndrome [27], and laminitis [28], as well as changes in the composition [29] and function [30] of the gut microbiota. Obesity can also negatively impact sports performances [31], as well as the physiological response to exercise and locomotion symmetry [32]. Moreover, an American survey showed that over-conditioned horses cost an average of $434 more per year for health as well as weight management tools than non-obese animals [33]. These findings highlight the direct detrimental effects of obesity on horses’ health and well-being, as well as the equine economy. 

In several species, maternal obesity has been shown to affect the health and behaviour of offspring through mechanisms associated with chronic inflammation in the mother and placenta [34,35,36]. For example, offspring from obese mothers have been shown to present an altered glucose metabolism [37] and cardiovascular function [38], an increased fat storage capacity in adipose tissue [39], as well as increased anxiety and altered food behaviour [40,41]. The present study follows previous research investigating the effect of maternal obesity on the development and growth of foals up to 18 months of age [17]. Despite receiving the same amount of energy based on body weight, Obese mares (group O) maintained a consistently high body condition score (BCS) (mean 4.0, 0–5 scale) during gestation, while Normal broodmares (group N) experienced a decrease in BCS during gestation (mean 3.1) and further reduction in BCS at the end of pregnancy, a period when foetal nutritional demands rise significantly. During gestation, O mares showed alterations in the adipose tissue endocrine system and increased low-grade inflammation. At the end of gestation, O mares were more insulin-resistant than N mares and had greater glucose effectiveness. Following birth, O foals exhibited an increased concentration of low-grade inflammation marker (Serum amyloid A) until 6 months of age and were more insulin resistant at 6 and 18 months of age, with a decreased glucose effectiveness compared to N foals. Finally, at 12 months of age, more O foals were affected by osteochondrosis lesions than N foals. 

We then demonstrated that maternal obesity affects the horse’s long-term health and development, as observed in other species. Placenta and milk, being both at the interface between the mother and the offspring, might be involved in maternal programming. In the horse, other factors, such as maternal nutrition, age, parity, nursing status, or breed, have been shown to affect placental structure and function as well as milk composition [4,42,43]. The effects of maternal obesity on placental structure and function as well as milk composition are, however, still unknown in horses.

The aim of this study was to compare the structure and function of the placenta at term and to discuss these results with respect to the fatty acid composition of the plasma, colostrum, and milk of obese and normal mares, as well as the plasma of their foals from birth up to 90 days of lactation. Based on the results observed in growing foals and in other species, we hypothesise that placentas from Obese mares will be more inflamed and have an increased expression of glucose transporters compared to Normal placentas. We also hypothesise that the milk of Obese mares will have a more pro-inflammatory fatty acid profile (i.e., increased content of omega-6 fatty acids and decreased content of omega-3 fatty acids). 

## 2. Materials and Methods

### 2.1. Ethical Statement

The animal studies were approved by the local animal care and use committee (“*Comité des Utilisateurs de la Station Expérimentale de Chamberet*”) and received ethical approval from the local ethics committee (« *Comité Régional d’Ethique pour l’Expérimentation Animale du Limousin* ») under protocol number 5-2013-5. 

### 2.2. Experimental Design, Management, and Feeding of Mares and Foals

The management of broodmares and foals, as well as dietary analyses, were previously described by Robles et al., 2018 [17]. Briefly, 24 multiparous dams were inseminated with the same stallion (French Anglo-Arab and *Selle Français* breeds). The mares were separated into 2 groups: 10 mares (group Normal (N)) with a mean BCS of 3 the year before insemination, and 14 mares (group Obese (O)) with a mean BCS of 3.8 the year before insemination [17]. The body condition of the mares was measured on a 0–5 scale. From insemination until 6 months of gestation, pregnant mares were housed all together in a single herd of grazing horses. Starting from the 6th month of gestation, they were housed individually in boxes (straw bedding) and fed the same amount of energy, protein, and fibres based on their body weight. Three days after foaling, mares and foals returned to pasture, managed in the same pastures until the foals were weaned at 6 months, with free access to water and a vitamin and mineral supplement. During the summer, specifically between the 2nd and 3rd months of lactation, the animals were given hay in addition to grazing due to a heatwave and drought (see supplementary data from [17] for more details). 

### 2.3. Measurements and Sampling 

#### 2.3.1. Measurements and Sampling at Foaling

Before the first suckling, jugular vein blood samples were collected from foals and mares using EDTA-coated tubes (Vacutainer, BD, Franklin Lakes, NJ, USA). Additionally, a 6 ml colostrum sample was collected from mares, and foals were weighed. Plasma was separated after a 10 min centrifugation at 5500 rpm at +4 °C. Subsequently, both the plasma and colostrum samples were stored at −20 °C. 

Immediately after delivery, placentas (allantochorion, without amnion) were weighted. For the measurement of placental surface area, allantochorions were placed in an “F” configuration, revealing both the uterine horns and the uterine body under a sheet of clear, transparent Plexiglas marked with 10 cm × 10 cm squares and photographed. Gross placental surface area was subsequently measured on the photographs using the ImageJ^®^ software (version 1.8.0_345, National Institute of Health, Bethesda, MD, USA) as described previously [44]. Placental volume was evaluated using a graduated water container by measuring the volume of water displaced after immersing the placenta into the water.

Placental samples, including the chorionic villi and the underlaying allantoic membrane, were collected within 30 min of delivery above the umbilical cord insertion, as described previously [45]. For each procedure, three aliquots were prepared: one set was fixed in 4% formaldehyde for histological analysis (1 cm² samples), while another set was snap-frozen (3 mm^2^ samples) and stored at −80 °C for gene expression analysis.

#### 2.3.2. Measurements and Sampling during Lactation

Milk samples were collected from mares at 30 and 90 days of lactation. For milk sampling, foals were muzzled for 3 h. After this period, both udders were completely emptied using a manual milker, as previously described [46]. Jugular vein blood samples were also collected into EDTA-coated tubes (Vacutainer, BD, Franklin Lakes, NJ, USA) at 30 and 90 days of lactation, following a 3 h fasting period, from both mares and foals. After manual homogenisation, approximately 10 mL of milk was retained, and both plasma and milk samples were stored at −20 °C.

### 2.4. Placental Analyses

#### 2.4.1. Histology and Stereology

Placental samples were embedded in paraffin. Sections (7 μm) were stained with hematoxylin/eosin using an automat (Varistain, Thermofisher, Waltham, MA, USA) for stereological analysis and scanned using NanoZoomer Digital Pathology^®^ (Hamamatsu Photonics, Hamamatsu, Japan). Surface densities (Sv) and Volume fractions (Vv) of the different components of the allantochorion, including microcotyledons and allantois, were quantified using One stop stereology with the Mercator^®^ software (ExploraNova, version 7.9.8, La Rochelle, France [47]). The measured components included haemotrophic and histiotrophic trophoblasts, microcotyledonary and allantoic vessels, microcotyledonary and allantoic connective tissues, and microcotyledons as the sum of all microcotyledonary components. The chorionic mesoderm and allantoic connective tissue were combined as allantoic connective tissue. Vv and Sv were then multiplied by the total volume of the placenta to estimate the absolute volume (cm^3^) and surface area (cm^2^) of the components of the allantochorion, as previously described [45].

For each placenta, 12 allantoic arterioles were randomly selected. The surface area of the lumen (calculated using transverse diameters) and the thickness of the vascular wall were measured using the ImageJ^®^ software (version 1.8.0_345, National Institute of Health, Bethesda, MD, USA).

#### 2.4.2. Functional Analyses

Total RNA from placental samples was isolated as described previously [48] and purified using an RNeasy Mini kit 250 (Qiagen, Hilden, NRW, Germany) following the manufacturer’s instructions. Reverse transcription and real-time quantitative PCR were performed as previously described [49]. Gene-specific primers for the analysed genes, involved in placental vascularisation, nutrient transport, and growth and development, are listed in Table 1.

### 2.5. Fatty Acid Analyses

To determine the fatty acid profiling and concentration, a known quantity of internal standard (margaric acid, C17:0) was added to 200 µL of full-fat milk and plasma before lipid extraction with chloroform/methanol (2:1, adapted from the method described by Folch et al. [50]). The fatty acids were transmethylated with 7% boron trifluoride methanol (Sigma-Aldrich, Saint Quentin Fallavier, France) in accordance with the method described previously by Morrison and Smith [51]. Finally, the methyl esters of milk fatty acids were analysed using gas chromatography (Auto Sampling 8410 Gas Chromatograph 3900, Varian, Les Ulis, France) coupled to a flame ionisation detector on an Econo-Cap EC-WAX capillary column (30 m, 0.32 mm internal diameter, 0.25 μm film, reference 19654, ALLTECH Associates Inc., Templemars, France), as described by Rousseau et al. [52]. Identification of fatty acids was made in reference to known fatty acid profiles obtained from the injection of a standard FAME (fatty acid methyl esters) mix (Supelco 37-component FAME mix, ref 47885-U, Sigma-Aldrich, Saint-Louis, MI, USA). The fatty acid profile was established for each sample and expressed as the percentage of total fatty acids.

Finally, 20 fatty acids were measured in milk and plasma from mares and plasma from foals, as presented in Table 2.

### 2.6. Statistical Analyses

Results are expressed as median [quartile 1–quartile 3] and presented as curves (median and interquartile range IQR) or boxplots (minimum to maximum). Statistical analyses and figures were performed using the *Rstudio* software (version 1.1.463., R version 3.5.1). For all analyses, effects were considered significant when *p*-values < 0.05 (or adjusted *p*-values < 0.05).

The results of placental biometry, stereology, and gene expression were analysed with a non-parametric permutation ANOVA (package lmPerm, function aovp) considering the group (N or O), the age, and wither’s height of the mare. For reasons of reproducibility, the seed was set at 1234. The sex of the foal had no effect on the measured parameters and was therefore not included in the model. Due to multiple testing for stereology and gene expression analyses, *p*-values were corrected using the fdr method.

For fatty acid profiles, two methods were used to analyse the data:A principal component analysis (PCA) was used for each point of measurement separately using the FactoMineR package.Non-parametric permutations ANOVA (package lmPerm, function aovp) considering the group (N or O), the age of the mare for plasma of mares and milk, and the sex of the foal for plasma of foals. The seed was set at 1234, and *p*-values were corrected using the fdr method. A volcano plot was drawn using the log2(fold-change) and the −10log(adjusted *p*-value) to summarise the linear model results. Boxplots and volcano plots were drawn using the ggplot2 package.

Missing data (between 4% and 39% depending on the sample type and time of sampling) were imputed using the missMDA package before analysis. The function imputeMultilevel was used to specify the data structure for the imputation.

PCA is a factorial method that enables the simultaneous study of several variables defined for the same set of individuals. Briefly, this method allows description, summarization, and reduction of the dimensionality of a dataset; it simplifies a multidimensional dataset into less numerous new variables called principal components. These principal components are therefore used to visualise the data graphically:–The individual factor map is a plot of the Principal Component Scores for individuals on the first two principal components. Confidence ellipses (0.95, around the barycenter) were added to the figure. A permutational multivariate analysis of variance (PERMANOVA, package vegan, function adonis2) was used to evaluate the difference between the barycenters of both groups, which was considered significant when *p* < 0.05.–The variable factor map presents a view of the projection of the observed variables projected into the plane spanned by the first two principal components. Also named a Correlation circle, it can help to visualise the most correlated variables (i.e., variables that group together) and to show the structural relationship between the variables and the components.–The projection of a variable vector onto the component axis allows us to directly read the correlation between the variable and the component. To simplify the reading of the figures, we merged the individuals and the variable factor maps into one biplot using the factoextra package (function fviz_pca_biplot).

Fatty acid concentrations were considered variables, while group and sampling type were considered qualitative factors. Total SFA: saturated fatty acids, MUFA: monounsaturated fatty acids, PUFA: polyunsaturated fatty acids, medium chain SFA (C10-14), medium chain MUFA (C10-C14), long chain SFA (C15-C20), long chain MUFA (C15-C20), ω-7 fatty acids, ω-9 fatty acids, ω-6 PUFA, ω-3 PUFA, ω3/ω6 PUFA ratio were added as supplementary variables. Individuals and variables were graphically represented on the two first dimensions.

Milk and plasma of mares and foals from 30 and 90 days of lactation were further analysed together to study the evolution of both samples during lactation using multifactorial analyses (MFA). PCA coordinates of each individual on the first and second dimensions were extracted, and Euclidean distances were calculated between samples: plasma of mares vs. milk, plasma of mares vs. plasma of foals, milk vs. plasma of foals. Results were analysed using a repeated-measures ANOVA with permutations using the aovp function (LmPerm package). For differences in fatty acid composition, Time and maternal age were considered fixed effects, while for distances, Time, Group, and their interaction were considered fixed effects. Individuals were considered a random effect in all models. Multiple comparisons were analysed with a permutation Tukey test (followed by a *p*-value correction using the fdr method), using the emmeans (emmeans package) and bootstrap_parameters (parameters package) functions.

Docosapentaenoic acid (DPA, C22:5ω3) was undetected in most samples and was therefore not included in the analyses, except for colostrum.

## 3. Results

### 3.1. Placental Analyses

#### 3.1.1. Neonate and Placental Biometry

There were no differences between groups for placental weight, surface and volume, foal birth weight, or placental efficiency. Maternal age was positively correlated with placental area (adjusted r2 = 0.25, β = 0.46, *p* = 0.03). Details are presented in Appendix A.

#### 3.1.2. Placental Stereology

No differences were observed between groups after *p*-value correction. In addition, no differences were observed for arteriolar vascular wall thickness or arteriolar lumen surface area. Details are presented in Appendix A.

#### 3.1.3. Placental Gene Expression

There was no effect of groups on placental gene expression at term. Expression of SLC2A3 (adjusted r2 = 0.446, β = 0.294, *p* = 0.037) increased with maternal age. Details are presented in Appendix A.

### 3.2. Fatty Acid Concentrations

There were no differences between groups for fatty acid concentration measured at any time in the plasma of mare plasma, foal plasma, or milk. In addition, obesity had no effect on colostrum IgG concentrations. Details are presented in Appendix A.

#### 3.2.1. Fatty Acid Composition in Plasma of Mares

PCA analysis could not separate the two groups of mares from birth until 3 months of lactation based on fatty acid composition. These results were confirmed using linear models. Figures and Tables associated with these results are presented in Appendix A.

#### 3.2.2. Fatty Acid Composition in Milk

Details associated with these results are presented in Appendix A.

##### Colostrum

Dimensions 1 (40.7%) and 2 (15.1%) of the PCA explained 55.8% of the total variance (Figure 1A). Normal and Obese groups were separated on the first dimension (r2 = 0.32, *p* = 0.004). The PERMANOVA confirmed that both groups were overall different (*p* = 0.005). After fdr correction, the colostrum of O mare was poorer in saturated fatty acids (SFA, *p* = 0.007, log2(FC) = −0.17) and especially in medium-chain saturated fatty acids (MC-SFA, *p* = 0.007, log2(FC) = −0.94) such as capric acid (C10:0, *p* < 0.0001, log2(FC) = −4.17) and lauric acid (C12:0, *p* = 0.013, log2(FC) = −1.85) (Figure 1B). Moreover, O colostrum was poorer in medium-chain monounsaturated fatty acids (MC-MUFA, *p* = 0.007, log2(FC) = −0.57) such as caproleic acid (C10:1, *p* < 0.0001, log2(FC) = −1.96) and lauroleic acid (C12:1ω9, *p* = 0.014, log2(FC) = −0.73), but richer in long-chain monounsaturated fatty acids (LC-MUFA, *p* = 0.018, log2(FC) = 0.20) such as palmitoleic acid (C16:1ω7, *p* = 0.025, log2(FC) = 0.31), hypogeic acid (C16:1ω9, *p* = 0.039, log2(FC) = 0.20), vaccenic acid (C18:1ω7, *p* = 0.006, log2(FC) = 0.32) and oleic acid (C18:1ω9, *p* = 0.066, log2(FC) = 0.15). Finally, Obese mares had milk richer in omega-6 fatty acids (*p* = 0.007, log2(FC) = 0.21), such as linoleic acid (LA, C18:2ω6, *p* = 0.013, log2(FC) = 0.21), but there was no difference in ω3/ω6 ratio between groups (Figure 1C).

##### At 30 Days of Lactation

No difference in milk fatty acid composition was observed between groups at 30 days of lactation. Figures and Tables associated with these results are presented in Appendix A.

##### At 90 Days of Lactation

The dimensions 1 (28.7%) and 2 (16.3%) of the PCA explained 45% of the total variance (Figure 2A). Normal and obese groups were separated by the second dimension (r2 = 0.39, *p* = 0.001). The PERMANOVA also showed a significant difference between groups (*p* = 0.025). Using the linear model, however, after fdr correction, only capric acid (C10:0, *p* < 0.0001, log2(FC) = −3.48) was significantly reduced in O compared to N milk using the permutation ANOVA (Figure 2B,C).

In summary, the colostrum of O mares was richer in long-chain monounsaturated fatty acids (LC-MUFA) and omega-6 PUFA (linoleic acid) and poorer in medium-chain saturated (MC-SFA) and monounsaturated (MC-MUFA) fatty acids. No differences were observed in milk fatty acid composition at 30 days of lactation. At 90 days of lactation, both milks started to separate again, with the milk of N mares being significantly richer in capric acid (C10:0) compared to the milk of O mares.

#### 3.2.3. Fatty Acid Composition in Plasma of Foals

Details associated with these results are presented in Appendix A.

##### At Birth

The PCA did not separate both groups of foals at birth and did the PERMANOVA test (*p* = 0.10) (Figure 3A). In details, however, O foals presented a reduced proportion of omega-3 fatty acids (*p* = 0.0096, log2(FC) = −0.44) as well as a reduced omega-3/omega-6 ratio (*p* = 0.007, log2(FC) = −0.50) in their plasma compared to N foals (Figure 3B). After fdr correction, ALA (C18:3ω3, *p* = 0.162, log2(FC) = −0.76) and EPA (C20:5ω3, *p* = 0.164, log2(FC) = −0.28) proportions were not statistically different, but their *p*-values were <0.05 before correction (*p* = 0.017 and *p* = 0.045, respectively). Moreover, DPA (C22:5ω3) was detected in the plasma of N foals only. ETA (C20:3ω3) was not included in the analysis as it was undetected in most samples (21 samples, over 24). Plasma proportions of ALA and EPA were reduced in foal plasma (Figure 3C).

##### At 30 Days of Age

The dimensions 1 (25.7%) and 2 (19.1%) of the PCA explained 44.8% of the total variance (Figure 4A). Ellipses of both groups of foals were separated on the first dimension (r2 = 0.38, *p* = 0.001). This result was confirmed with the PERMANOVA (*p* = 0.016). Accordingly, after fdr correction, the plasma of O foals was poorer in medium-chain saturated fatty acids (MC-SFA, *p* < 0.0001, log2(FC) = −0.43), such as lauric acid (C12:0, *p* = 0.042, log2(FC) = −0.97) and myristic acid (C14:0, *p* < 0.0001, log2(FC) = −0.43) compared to N foals (Figure 4B). Differences in medium-chain mono-unsaturated fatty acids (MC-MUFA, *p* = 0.026, log2(FC) = −0.52) were also observed as caproleic acid (C10:1, *p* = 0.071, log2(FC) = −1.29) and lauroleic acid (C12:1ω9, *p* = 0.004, log2(FC) = −0.34) were impoverished in the plasma of O compared to N foals. Finally, plasma of O foals was also richer in palmitic acid (C16:0, *p* = 0.002, log2(FC) = −0.09) and gondoic acid (C20:1ω9, *p* = 0.003, log2(FC) = −0.58) (Figure 4C).

##### At 90 Days of Age

The first (29%) and second (15.4%) dimensions explained 44.4% of the total variance (Figure 5A). Ellipses of both groups of foals were separated on the first dimension (r2 = 0.65, *p* < 0.0001). This result was confirmed by the PERMANOVA (*p* = 0.001). Accordingly, after fdr correction, the plasma of O foals was poorer in medium-chain saturated fatty acids (MC-SFA, *p* < 0.0001, log2(FC) = −0.48) such as lauric acid (C12:0, *p* < 0.0001, log2(FC) = −0.91), myristic acid (C14:0, *p* < 0.0001, log2(FC) = −0.51), and pentadecylic acid (C15:0, *p* = 0.049, log2(FC) = −0.19) (Figure 5B). Conversely, O foals had a plasma richer in long-chain saturated fatty acids (LC-SFA, *p* < 0.0001, log2(FC) = 0.07) such as C18:0 (*p* = 0.025, log2(FC) = 0.12). O foals had a plasma poorer in medium-chain monounsaturated fatty acids (MC-MUFA) such as 3-decylenic acid (C10:1ω7, *p* = 0.004, log2(FC) = −0.87) and pentadecenoic acid (C15:1ω9, *p* = 0.003, log2(FC) = −0.47) but richer in vaccenic acid (C18:1ω7, *p* < 0.0001, log2(FC) = 0.23). Finally, foals born to obese mares had a plasma poorer in omega-3 fatty acids (*p* = 0.004, log2(FC) = −0.20) with a lower ω3/ω6 ratio (*p* = 0.001, log2(FC) = −0.31) and reduced ALA (C18:3ω3, *p* = 0.024, log2(FC) = −0.19) compared to N foals (Figure 5C).

In summary, no differences were observed in plasma fatty acid composition between both groups of foals at birth. At 30 days of age, O foals had a plasma low in saturated fatty acids and omega-9 monounsaturated fatty acids compared to N foals. At 90 days of age, O foals had a plasma poorer in medium-chain saturated (MC-SFA) and monounsaturated (MC-MUFA) fatty acids, as well as omega-3 fatty acids, but richer in stearic acid (C18:0) and vaccenic acid (C18:1ω7).

#### 3.2.4. Comparison of Samples during Lactation

##### Euclidean Distances

No differences were observed at 30 and 90 days of lactation for the distances between plasma from mares and plasma from foals (Figure 6A). At 30 days of lactation, the distance of the fatty acid profile between plasma from mares and milk was not different between the two groups (Figure 6B). At 90 days of lactation, however, the distance between plasma and milk increased in N mares (N 30d vs. N 90d, *p* = 0.004) but not in O mares (O 30d vs. O 90d, *p* = 0.68). As a result, the distance between plasma and milk was lower in O mares at 90 days of lactation compared to N mares (*p* = 0.016). At 30 days of lactation, no differences were observed between groups for the distances between plasma from foals and milk (Figure 6C). At 90 days of lactation, however, the distance between plasma and milk increased in N foals (N 30d vs. N 90d, *p* = 0.012) but not in O foals (O 30d vs. O 90d, *p* = 0.36). As a result, the distance between plasma and milk was lower in O foals at 90 days of lactation compared to N foals (*p* = 0.018).

##### Effect of Time of Lactation on Plasma Fatty Acid Profiles from Mares

With few exceptions, changes in plasma fatty acid abundance were similar in both groups of mares between 30 and 90 days of lactation (Figure 7). Proportions of medium-chain saturated fatty acids (MC-SFA, N, *p* < 0.0001, log2(FC) = −0.70, O, *p* < 0.0001, log2(FC) = −0.54), as well as palmitic acid (C16:0), were decreased in the Normal group (C10:0, *p* < 0.0001, log2(FC) = −1.78, C12:0, *p* = 0.003, log2(FC) = −0.94, C14:0, *p* < 0.0001, log2(FC) = −0.79, C16:0, *p* < 0.0001, log2(FC) = −0.16) and the Obese group (C10:0, *p* < 0.0001, log2(FC) = −1.28, C12:0, *p* < 0.0001, log2(FC) = −0.84, C14:0, *p* < 0.0001, log2(FC) = −0.60, C16:0, *p* = 0.008, log2(FC) = −0.12). In both groups, stearic acid (C18:0) increased at 90 days compared to 30 days of lactation (N, *p* < 0.0001, log2(FC) = 0.21, O, *p* < 0.0001, log2(FC) = 0.21). Finally, omega-3 fatty acid abundance decreased (N, *p* = 0.011, log2(FC) = −0.35, O, *p* = 0.0008, log2(FC) = −0.25) while omega-6 fatty acids increased (N, *p* = 0.014, log2(FC) = 0.19, O, *p* = 0.002, log2(FC) = 0.17), leading to a reduced omega-3/omega-6 ratio in both groups (N, *p* = 0.001, log2(FC) = −0.45, O, *p* < 0.0001, log2(FC) = −0.43).

##### Effect of Time of Lactation on Milk Fatty Acid Profiles

The proportion of lauric acid (C12:0) was reduced (N, *p* < 0.0001, log2(FC) = −1.32, O, *p* < 0.0001, log2(FC) = −1.12), and the proportion of palmitoleic acid (C16:1ω7) was increased (N, *p* < 0.0001, log2(FC) = 0.32, O, *p* = 0.02, log2(FC) = 0.30) at 90 days of lactation compared to 30 days lactation in both groups of mares (Figure 8). In Obese mares, capric acid abundance was also strongly reduced at 90 days of lactation (*p* = 0.004, log2(FC) = 3.41), resulting in a decrease of MC-SFA (*p* < 0.0001, log2(FC) = −0.52). Conversely, the abundance of omega-9 fatty acids (*p* = 0.003, log2(FC) = 0.14) increased at 90 days of gestation, such as oleic acid (C18:1ω9, *p* = 0.042, log2(FC) = 0.13) and gondoic acid (C20:1ω9, *p* = 0.023, log2(FC) = 0.56) compared to 30 days of gestation in Obese milk.

##### Effect of Time of Lactation on Plasma Fatty Acid Profiles from Foals

In both groups of foals, pentadecylic acid (C15:0, N, *p* < 0.0001, log2(FC) = 0.49, O, *p* < 0.0001 log2(FC) = 0.24), myristoleic acid (C14:1ω9, N, *p* < 0.0001, log2(FC) = 0.92, O, *p* < 0.0001, log2(FC) = 1.03) and eicosapentaenoic acid (EPA, C20:5ω3, N, *p* < 0.0001, log2(FC) = 1.67, O, *p* < 0.0001, log2(FC) = 1.47) proportions were higher at 90 compared to 30 days of age (Figure 9). Conversely, abundance of oleic acid (C18:1ω9) was reduced at 90 days of age in foals born to Normal (*p* = 0.033, log2(FC) = −0.13) and Obese mares (*p* = 0.005, log2(FC) = −0.14). In Normal group only, lauric acid proportion increased (C12:0, *p* = 0.01, log2(FC) = 0.69) while palmitic abundance decreased (C16:0, *p* < 0.0001, log2(FC) = −0.15) between 30 and 90 days of age. In Obese group only, the gondoic acid proportion was reduced at 90 days of age compared to 30 days of age (C20:1ω9, *p* = 0.042, log2(FC) = 0.22).

In summary, differences in fatty acid profiles in plasma from mares occurring between 30 and 90 days of lactation were similar between groups. In O milks, the abundance of capric acid was reduced, while omega-9 fatty acids were more abundant at 90 days of lactation. These differences were not observed in N milks. In plasma from foals, lauric acid proportion increased and palmitic acid abundance decreased in the N group, while gondoic acid proportion decreased at 90 days of age in the O group.

## 4. Discussion

The present study investigated the effect of maternal obesity during gestation and lactation on placental morphometry, gene expression, and structure at term, as well as fatty acid profiles in plasma from mares, foals, and milk up to 3 months of lactation. Overall, maternal obesity did not affect the placental variables measured; placentas exhibited similarity in size, weight, volume, and efficiency, with no differences in placental structure or gene expression.

Fatty acid profiles in the plasma of mares did not differ between the groups. At birth, however, colostrum samples exhibited different fatty acid profiles between groups, with an increased proportion of long-chain unsaturated fatty acids in O samples. These differences were no longer observed at 30 days of lactation. Still, the milk from Normal and Obese mares exhibited differences again at 90 days of lactation, with a decreased proportion of capric acid in the milk of O mares. Moreover, the plasma from foals presented different fatty acid profiles from birth until 90 days of age. The differences between plasma from mares and milk as well as plasma from mares and foals increased during lactation for N mares but not for O mares.

### 4.1. Placental Measurements and Plasma from Foals at Birth

No effect of maternal obesity was observed on foal birthweight or placental morphometry. These results are not surprising, as only extreme pathologies (such as extreme undernutrition or bacterial infection) have been shown to affect feto-placental measurements at birth in horses [2,43].

After *p*-value correction, no difference in placental structure or gene expression was observed. In humans and animal models, maternal obesity is primarily associated with alterations in lipid metabolism resulting in altered expression and/or activity of fatty acids [53,54,55], glucose and amino acid transporters [56], placental lipid accumulation (sex-specific effects), decreased fatty acid oxidation [57,58,59,60], the development of a lipotoxic environment (production of oxidised lipids), and reticulum endoplasmic stress due to oxidative stress [58,61,62]. Maternal obesity has also been shown to be associated with significant activation of inflammation systems and impaired vascular development and function [63,64,65,66,67]. A thickening of the vascular wall is indeed observed in cases of maternal obesity or insulin resistance (gestational diabetes) [65,68]. This thickening corresponds to an increased development of smooth muscles and perivascular fibrosis, which leads to a decrease in vessel elasticity. In mice, maternal obesity has been shown to be associated with a decreased placental oxygen diffusing capacity [58]. In the horse, placentas of insulin-resistant mares (suffering from laminitis and fed with cereals in late gestation) also present vascular wall vessel thickening in the allantoic arterioles and an altered expression of genes involved in inflammation and vascular development [45,69]. An increased expression of genes involved in inflammation has also been observed in the endometrium of obese mares and the trophectoderm of their embryos at 16 days of gestation [70]. The vasodilatation ability of placentas from obese dams may then be reduced, as observed in vivo by laser velocimetry in women [63]. The mares in the present study were fed the same quality and quantity of cereals during late gestation as the mares in a previous study [45]. When comparing the placental vascular measurements of both studies, the placentas of Obese mares, Normal mares, and mares fed with cereals in late gestation presented similar arteriole vascular wall thickness and lumen surface area that were altered compared to mares fed with forages only. Our hypothesis is that the diet of mares affected the structure of the allantoic arterioles, with a stronger effect than obesity. Moreover, placental alterations due to maternal obesity in humans are likely to be the result of both maternal nutrition and metabolic alterations.

At birth, there was no difference in plasma fatty acid profiles in mares. O foals, however, presented already at birth a reduced plasma omega-3/omega-6 ratio, associated with an increased proportion of omega-3 fatty acids. Omega-6 and omega-3 fatty acid precursors, namely linoleic (LA) and alpha-linolenic acids (ALA), respectively, are essential fatty acids that mammals cannot synthesise and must find in their dietary intake [71]. Essential plasma fatty acid concentrations reflect not only the nutritional state of the individual [72,73], but also their metabolism, as C18 PUFA can be elongated and desaturated further. ALA can be metabolised into eicosapentaenoic (EPA, C20:5ω3) and docosahexaenoic (DHA, C22:6ω3) acids, which are precursors of anti-inflammatory, vasodilator, and anti-coagulant factors [74]. The difference in plasma fatty acids from foals may be a consequence of maternal metabolic status-induced modification of the quantity and/or activity of placental fatty acid transporters, but also of placental lipid metabolism. Maternal obesity had no effect on placental gene expression (transcriptomic level) of *lipoprotein lipase* (*LPL*) or *cluster of differentiation 36* (*CD36*) in the present study, but expression of other genes involved in fatty acid transport and metabolism, as well as protein quantity, transporter activity, and tissue-specific gene expression, have not been studied. The effects of obesity on LPL and other fatty acid transporters are inconsistent based on the model studied, the diet, and the time of gestation [54,55]. Although it has been shown in the horse that there was an increased gene expression of *fatty acid synthase* (*FASN*) and a decreased gene expression of *CD36* in the trophectoderm of embryos of obese mares at 16 days of gestation [70], as we only studied placental gene expression at term, alterations in gene expression may have occurred earlier in pregnancy and may have affected placental function and foal development. In humans, obesity has also been associated with a decreased placental proportion of EPA and DHA and altered expression of genes and proteins, indicating a decreased placental DHA transfer capacity in male foetuses [59]. Fatty acid binding protein 3 (FABP3) has also been stipulated to be the main actor in the omega-3 PUFA transfer in the placenta [75]. In humans, the ω-3/ω-6 PUFA ratio has been shown to be decreased in the umbilical artery of women with gestational diabetes at birth (insulin resistance), which is corroborated by our results [76]. Obese mares presented an increased concentration of serum amyloid A in late gestation, indicating increased systemic inflammation [17]. These alterations may also have affected placental lipid metabolism and lipid transport. These differences indicate alterations of placental function in obese gestations that may have been involved in the programming of insulin homeostasis as well as systemic inflammation and osteo-articular lesions observed in obese foals later during growth.

The effects of maternal obesity on placental function remain to be studied in the horse to better understand the role of this organ in equine DOHaD.

### 4.2. Fatty Acid Profiles

#### 4.2.1. Plasma from Mares

We previously observed in these animals that obese mares tended to have an increased plasma concentration of serum amyloid A at foaling [17], which is considered a marker of systemic inflammation. Moreover, these mares had reduced insulin sensitivity compared to normal mares, which is commonly associated with systemic inflammation in the horse. We expected differences in fatty acid composition between both groups of mares, with an increased abundance of omega-6 fatty acids and/or a decreased abundance of omega-3 fatty acids in the plasma of obese mares. Differences in lipid composition were observed between horses with metabolic syndrome and control horses in a previous study [77]. Lipidomics, however, using mass spectrometry was used to detect, for example, mono-, di-, and triacylglycerols, ceramides, and sphyngomyelins in the plasma of horses while we analysed the profile of total fatty acids in the present study. Another reason for this lack of difference would be that both groups of mares had the same diet during pregnancy (hay, haylage, and barley) and lactation (housed in the same pasture and hay). Diet is a factor known to strongly affect the plasma fatty acid profile in equids [73,78] and other species [79].

#### 4.2.2. Colostrum

In the present study, O colostrum was poorer in medium-chain saturated fatty acids (MC-SFA) than N colostrum. As MC-SFA is an important component of milk in horses compared to humans [80], it likely plays an important role in foal nutrition. MC-SFA is quickly hydrolysed (no need for bile acid action) and absorbed (free diffusion) in the intestine of monogastrics [74]. These fatty acids are efficiently used as an energy source by the liver; as carnitine palmitoyltransferase is not needed for their transfer into the mitochondria, they are therefore considered an immediate energy source for the newborn [74]. Thus, newborn piglets have been shown to be more efficient at oxidising MC-SFA than LC-SFA [81]. Moreover, MC-SFA has been shown to improve the intestinal morphology in rats [82] and pigs [83] and to modulate the immune system (increased secretion of immunoglobulin A (IgA) [84], of cytokines produced after Ig stimulation [85], and of anti-inflammatory cytokines; decreased secretion of pro-inflammatory cytokines [84]) after LPS-injection in the same species. Other studies also showed that MC-SFA can act as an anti-microbial agent and therefore play a role in the development of the microbiota in newborns [74]. In conclusion, foals born to Obese dams likely received less MC-SFA at birth, receiving less immediate energy as MC-SFA, which could also affect their intestinal immune system and microbiota; however, this did not affect the growth of foals as their growth rate between birth and 1 day of age and between birth and 7 days of age was not different between groups [17].

Conversely, Obese mares had colostrum richer in long-chain monounsaturated fatty acids (LC-MUFA) compared to normal mares. In humans, obese women produce colostrum that is poorer in long-chain saturated fatty acids (LC-SFA) such as palmitic (C16:0) and stearic (C18:0) acids and in omega-3 fatty acids such as ALA (C18:3ω3) compared to normal women [86,87,88]. Conversely, the colostrum of obese women is richer in long-chain monounsaturated fatty acids (LC-MUFA) such as C16:1ω7, C16:1ω9, and C18:1ω9 and omega-6 fatty acids such as LA (C18:2ω6) and AA (C20:4ω6) [86,87,88]. In both species, we therefore observe an increased proportion of LC-MUFA and omega-6 fatty acids in the colostrum of obese individuals. Oleic acid (C18:1ω9) is synthesised from stearic acid (C18:0). Palmitoleic acid (C16:1ω7) is synthesised from palmitic acid (C16:0) and has been shown to improve insulin sensitivity in the liver and muscles in rodents, humans, and sheep [89,90,91]. These two LC-MUFAs are principal components of triglycerides. In cows, long-chain fatty acids cannot be synthetized de novo by the mammary gland and therefore have to be absorbed from the systemic circulation, but desaturation of LC-SFA into LC-MUFA is catalysed by SCD1 in the mammary gland; however, to our knowledge, this information is not known in equids. Linoleic acid is an omega-6 polyunsaturated fatty acid that can be metabolised into arachidonic acid (AA) by mammals, including horses. Here, colostrum from obese mares was richer in LA but not in AA. No differences in the fatty acid profile of plasma from mares were observed at birth, reflecting mainly the forages and concentrates consumed. If, as in cows, long-chain fatty acids cannot be synthesised in the mammary gland of mares, the differences in colostrum composition between obese and normal mares may be explained by:–An increase in SCD1 expression and/or activity in colostrum from obese mares. In goat and buffalo mammary epithelial cells in vitro, an overexpression of SCD1 leads to an increased production of oleic and palmitoleic acids [92]. We showed previously that O mares were insulin resistant in late gestation [17]. In bovine mammary epithelial cells in vitro, insulin was shown to increase SCD1 promoter activity [93], and SCD1 activity was shown to be increased in insulin-resistant individuals in humans and rodents [94]. These differences in glucose metabolism may therefore affect the expression and/or activity of SCD1 in the mammary glands of obese mares.–An increase in long-chain fatty acid transport in the mammary gland. PPARG1 has been shown in goats to upregulate the expression of LPL and CD36, which are involved in long-chain fatty acid transport [95]. However, there is no information on the effect of obesity on the mammary gland gene expression of SCD1, LPL, CD36, and PPARG1. More research is needed to understand the physiology of the mammary gland in equids and the effect of obesity on mammary gland gene and protein expression.

Linoleic acid (LA, C18:2ω6) is an essential fatty acid from the omega-6 family. A horse cannot synthesise this fatty acid, meaning that diet is the sole source of LA for pregnant and lactating mares. As for LC-SFA and LC-MUFA, the differences in LA abundance between the colostrum of both groups may therefore be associated with an increased transport of long-chain fatty acids in the mammary glands of Obese mares. Interestingly, however, alpha-linolenic acid abundance did not differ between groups. Horses can further elongate and desaturate LA into arachidonic acid (AA, C20:4ω6). Arachidonic acid is a precursor of pro-inflammatory, vasoconstrictive, and pro-coagulant molecules [74].

#### 4.2.3. Milk and Plasma of Foals

In rodents, obesity is associated with a decreased expression and activity of acetyl-coA decarboxylase (ACC) at 10 days of lactation, indicating a decrease in fatty acid de novo synthesis [62,96]. These results are consistent with what we observe in the present study. Indeed, Euclidean distances calculated between the fatty acid profile of milk and the plasma of mares showed an increase at 90 days compared to 30 days of lactation in normal mares but not in obese mares. This indicates that, at 90 days of lactation, normal mares had a milk fatty acid composition significantly different from their plasma than obese mares which could imply a decreased fatty acid de novo synthesis in the mammary gland of obese mares. At 1 day of lactation, in rodents, the authors observed a decreased expression of ACC, but associated with an increased enzyme activity that led to a similar de novo fatty acid synthesis between groups [62,96]. Moreover, both groups of mares lost body condition during the summer. N mares reached their lowest body condition at a median of 2.25/5, while obese mares stayed significantly fatter at a median of 3.25/5 (although they were not considered overweight anymore at that time). There is therefore also a possibility that O mares were using their fat stores to synthesise milk fatty acids, while N mares mostly used fatty acids from their diet as their fat stores were low. Such adaptation could also explain why no differences were observed for milk fatty acid composition between groups of mares at 30 days of lactation in the present study. Interestingly, these differences in distances are also observed in foals, which was expected since foals are still drinking milk at 90 days of age, even though dietary diversification begins early in horses.

At 90 days of lactation, after fdr correction, only capric acid (C10:0) was significantly less abundant in the milk of O mares, even though the PCA analysis separated the two groups. Before the fdr (false discovery rate) correction (confirmed by the PCA), the milk of O mares was richer in lauric acid (C12:0), 3-decylenic-acid (C10:1), and lauroleic acid (C12:1) and poorer in palmitic acid (C16:0). A heatwave happened during the 2nd and 3rd months of lactation and can be considered a nutritional challenge (for energy, even if hay was added to pastures, but also for PUFA diet content as drought likely decreased the PUFA content in grass) and a heat stress for the herd. Obese mares may therefore have been less able to adapt to these changes than Normal mares, which could have affected the quality of their milk. In cows, obesity is associated with a reduced ability to adapt to heat stress, which leads to a greater decrease in milk production and total fat than in normal BCS cows [97]. Studies on the interaction between obesity and heat stress are scarce, and to our knowledge, milk fatty acid composition has not been studied. In normal BCS cows, heat stress is associated with a decrease of MC-SFA and MC-MUFA and an increase of C18 fatty acids (C18:0, C18:1ω9, and C18:2ω6) in milk [98]. In the present study, a decrease in MC-SFA proportion between 30 and 90 days of lactation is observed in Obese (C10:0 and C12:0) but not Normal milks (only C12:0 for N mares). Moreover, an increase in more LC-MUFA is observed in Obese milks (C16:1ω7, C18:1ω9, and C20:1ω9) compared to Normal milks (C16:1ω7). These results are consistent with what was observed in bovines and seem to indicate a stronger effect of heat stress on Obese mares compared to Normal mares. One hypothesis proposed in cows is that heat stress decreases de novo synthesis and increases long-chain fatty acid transport and/or desaturation in the mammary gland [98], effects that would have been increased in Obese mares as observed with Euclidean distances. With climate change, more studies are needed to understand the effect of heat stress on the development and milk quality of horses in interaction with maternal factors such as obesity and insulin resistance.

The fatty acid composition in the plasma of foals started to be different at 30 days of age, despite no difference in milk fatty acid profiles between groups. At 30 days of age, O foals had a plasma poorer in saturated fatty acids (C12:0, C14:0, and C16:0) as well as in mono-unsaturated fatty acids (C12:1 and C20:1ω9). These differences may be associated with changes in feeding behaviour, as foals start grazing very early after birth. At 1 month of age, grazing time reaches 80min/day for 69min/day of suckling [99]. The behaviour of foals was not measured in this study. In humans and macaques, offspring of obese or overnourished dams are more anxious than offspring of control dams [100,101]. Moreover, it has been shown in rats and sheep that offspring of overnourished mothers spend more time to suckle compared to offspring of control dams [102,103]. It could then be possible that O foals spent more time suckling than N foals, leading to a decreased time spent grazing and then to the difference in plasma fatty acid composition observed between groups.

Differences in foal metabolism could also explain these results. In rodents, maternal obesity during gestation has been shown to affect the hepatic mitochondrial function of the offspring [104], where fatty acid oxidation occurs. Rats born to insulin-resistant dams have an increased hepatic activity of SCD1, resulting in an increased proportion of C16:1ω7 lipids in the liver [105]. In humans, mesenchymal stem cells derived from the umbilical cord of infants born to obese mothers have an increased lipid content and altered fatty oxidation compared to normal mesenchymal stem cells [106]. These alterations in lipid metabolism are often associated with insulin resistance [105]. In the present article, the metabolism of foals was evaluated at 6, 12, and 18 months of age, and foals born to Obese mares had a reduced insulin sensitivity at 6 and 18 months of age compared to foals born to Normal mares. No conclusion can be drawn from the data available, and more research is needed to understand the biological mechanisms behind these differences. Another hypothesis would be a difference in fatty acid intestinal absorption between groups, but this remains to be studied.

At 90 days of age, foals born to Obese mares had a plasma richer in stearic acid (C18:0) as well as vaccenic acid (C18:1ω7), but poorer in MC-SFA (C12:0, C14:0, and C15:0), MC-MUFA (C10:1 and C15:1ω9), as well as ALA (C18:3ω3) with a decreased omega-3/omega-6 ratio. We showed that foals born to Obese mares had an increased concentration of serum amyloid A (a marker of systemic inflammation) at 90 days of age, which correlates with the pro-inflammatory profile observed in their plasma [17]. Interestingly, even though the milk of Obese mares was richer in lauric acid (C12:0) and 3-decelynic-acid (C10:1), these fatty acids were less abundant in the plasma of O foals. As presented above, differences in lipid metabolism (oxidation, desaturation, transport, and storage), digestion, absorption, or an interaction between maternal milk and adaptation to heat stress could explain these results, but more studies are needed to understand how maternal obesity affects the health and development of foals.

### 4.3. Study Limitations

This study has several limitations that may have affected the interpretation of the results:–Sample size: Ten mares were in the Obese group and 14 in the Normal group. This sample size is considered low and may have led to low statistical power (especially with the multiple test correction). This could have inflated the risk of false negatives. Therefore, this study needs to be replicated in order to confirm the absence of differences in placental structure and function between obese and non-obese mares, as well as differences in fatty acids in the analysed tissues between groups.–Pasture quality information: No information was available for pasture quality or pasture consumption by mares and foals during the lactation period. Therefore, the intensity of the drought could not be measured directly on the fields. The fact that, despite having unlimited access to hay, mares of both groups lost body condition during the heat stress episode highlighted the severity of this event.–Method of milk sampling: Milk was sampled after a 3 h waiting period, with foals muzzled to prevent them from suckling their mother. Our results may therefore differ from studies in which milk was sampled directly. This waiting period may have decreased the fatty acid concentration and altered the udder’s fatty acid metabolism. Because all mares and foals were sampled using the exact same protocol, our results are comparable to each other’s but may not be comparable with other studies.

### 4.4. Study Perspectives for the Equine Industry

In our initial study, we showed that foals born to obese mares were more susceptible to present osteo-articular lesions and had an altered glucose metabolism [17]. These effects of maternal obesity on the foals’ programming development may therefore be the combination of in utero factors (maternal inflammation and insulin resistance) as well as lactation factors (the pro-inflammatory profile of milk from obese mares). The effects of maternal obesity may therefore affect offspring health and performance in adulthood, which could have direct or indirect effects (monetary but also well-being) on the equine industry.

The milk of obese mares shows a decreased nutritional quality compared to normal mares for human consumption. Horse milk can be considered a substitute for cow-based milk for children with food allergies [107], which would therefore decrease the value of the product.

Moreover, in the present study, Obese mares showed a lower ability to adapt to heat stress compared to Normal mares. This inability to adapt to heatwave periods can further increase the detrimental effects of maternal obesity on the foals’ development. Effects on maternal health and fertility were not studied in this article, and to our knowledge, they are currently unknown in the horse species. Climate change is intensifying around the world, particularly in Europe. This continent is warming twice as fast as the rest of the world, and the occurrence of heatwave and drought events is already increasing above prediction thresholds [108]. These results identify obese mares as more susceptible to heatwaves’ detrimental effects and provide another argument for better monitoring of body condition in broodmares.

Finally, to better understand the effects of maternal obesity on the foal’s development, a study of the mothers and foals’ behaviour would add both knowledge on the mechanisms involved in programming the offspring phenotype as well as potential ways to improve breeding management of obese broodmares.

## 5. Conclusions

In conclusion (Figure 10), maternal obesity did not affect term placental morphometry, structure, or gene expression. Obesity did not affect the fatty acid composition in the plasma from mares during lactation but decreased the omega-3 proportion and the omega-3/omega-6 ratio in the plasma from foals. The fatty acid profile of colostrum from Obese mares nevertheless indicated differences in mammary gland metabolism (increased long-chain fatty acid transfer and desaturation) with those from Normal mares. Differences in milk fatty acid composition might be dependent on the ability of the mares to adapt to heat stress. Finally, the plasma fatty acid profile of foals born to Obese mares was more pro-inflammatory, which correlated with the increased systemic inflammation observed in this group. Further investigations are needed to understand the mechanisms explaining these results and to confirm our hypotheses.

## Figures and Tables

**Figure 1 vetsci-10-00691-f001:**
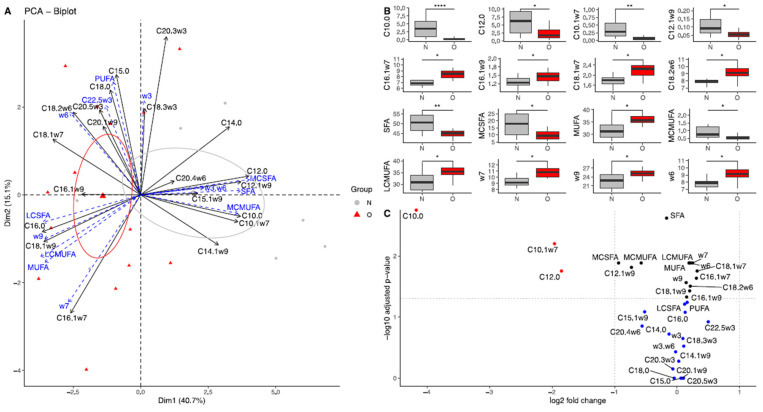
Fatty acid profile in colostrum from Normal mares (N, n = 14) and Obese mares (O, n = 10). (**A**) Biplot of principal component analysis, representing the first (40.7%) and second dimensions (15.1%). Variables included in the analysis are represented by the black arrows. Supplementary variables are represented by the blue arrows. Colostrum samples from Normal mares are represented by the grey points and ellipse, while colostrum samples from Obese mares are represented by the red points and ellipse. The ellipses represented are 95% confidence ellipses around the barycenter. (**B**) Box plot of fatty acids with a significant difference in abundance (% of total fatty acids) between Normal and Obese mares. * = *p* < 0.05, ** = *p* < 0.01, **** = *p* < 0.0001. (**C**) Volcano plot representing the log2(fold-change) and the -log10(pvalue). In blue are the fatty acids with no significant differences between groups; in black, the fatty acids are significantly different but have a log2(fold-change) between −1 and 1, and in red, the fatty acids are significantly different with a high fold-change. SFA = saturated fatty acids, MC-SFA = medium-chain saturated fatty acids, LC-SFA = long-chain saturated fatty acids, MUFA = monounsaturated fatty acids, MC-MUFA = medium-chain monounsaturated fatty acids, LC-MUFA = long-chain monounsaturated fatty acids, PUFA = polyunsaturated fatty acids, w3= omega-3 fatty acids, w6= omega-6 fatty acids, w3.w6 = omega-3/omega-6 fatty acids ratio.

**Figure 2 vetsci-10-00691-f002:**
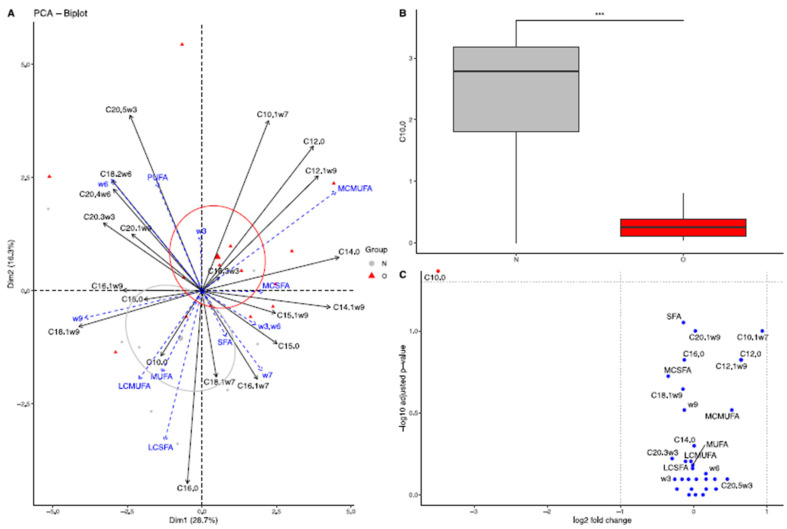
Fatty acid profile in milk from Normal mares (N, n = 14) and Obese mares (O, n = 10) at 90 days of lactation. (**A**) Biplot of principal component analysis representing the first (29.1%) and second dimensions (17%). Variables included in the analysis are represented by the black arrows. Supplementary variables are represented by the blue arrows. Milk samples from Normal mares are represented by the grey points and ellipse, while milk samples from Obese mares are represented by the red points and ellipse. The ellipses represented are 95% confidence ellipses around the barycenter. (**B**) Box plot of fatty acids with a significant difference in abundance (% of total fatty acids) between Normal and Obese mares. *** = *p* < 0.001. (**C**) Volcano plot representing the log2(fold-change) and the -log10(qvalue). In blue are the fatty acids with no significant differences between groups; in black, the fatty acids are significantly different but have a log2(fold-change) between −1 and 1, and in red, the fatty acids are significantly different with a high fold-change. SFA = saturated fatty acids, MC-SFA = medium-chain saturated fatty acids, LC-SFA = long-chain saturated fatty acids, MUFA = monounsaturated fatty acids, MC-MUFA = medium-chain monounsaturated fatty acids, LC-MUFA = long-chain monounsaturated fatty acids, PUFA = polyunsaturated fatty acids, w3 = omega-3 fatty acids, w6 = omega-6 fatty acids, w3.w6 = omega-3/omega-6 fatty acids ratio.

**Figure 3 vetsci-10-00691-f003:**
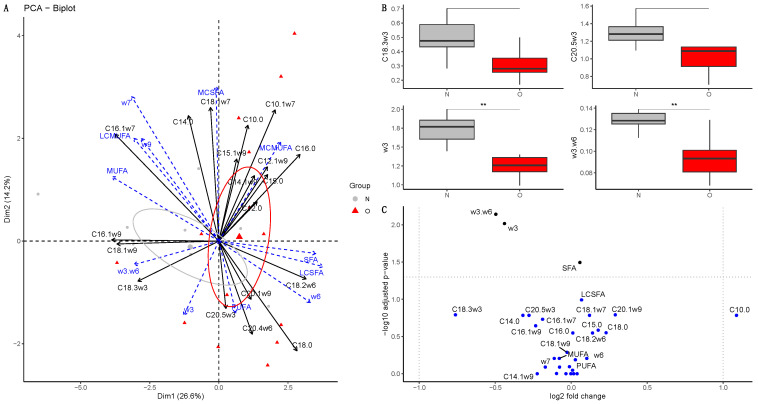
Fatty acid profile in the plasma from foals born to Normal mares (N, n = 14) and Obese mares (O, n = 10) at birth. (**A**) Biplot of principal component analysis representing the first (26.6%) and second dimensions (14.2%). Variables included in the analysis are represented by the black arrows. Supplementary variables are represented by the blue arrows. Plasma samples from Normal foals are represented by the grey points and ellipse, while plasma samples from Obese foals are represented by the red points and ellipse. The ellipses represented are 95% confidence ellipses around the barycenter. (**B**) Box plot of fatty acids with a significant difference in abundance (% of total fatty acids) between Normal and Obese foals. ** = *p* < 0.01. (**C**) Volcano plot representing the log2(fold-change) and the −log10(qvalue). In blue are the fatty acids with no significant differences between groups; in black, the fatty acids are significantly different but have a log2(fold-change) between −1 and 1, and in red, the fatty acids are significantly different with a high fold-change. SFA = saturated fatty acids, MC-SFA = medium-chain saturated fatty acids, LC-SFA = long-chain saturated fatty acids, MUFA = monounsaturated fatty acids, MC-MUFA = medium-chain monounsaturated fatty acids, LC-MUFA = long-chain monounsaturated fatty acids, PUFA = polyunsaturated fatty acids, w3= omega-3 fatty acids, w6= omega-6 fatty acids, w3.w6 = omega-3/omega-6 fatty acids ratio.

**Figure 4 vetsci-10-00691-f004:**
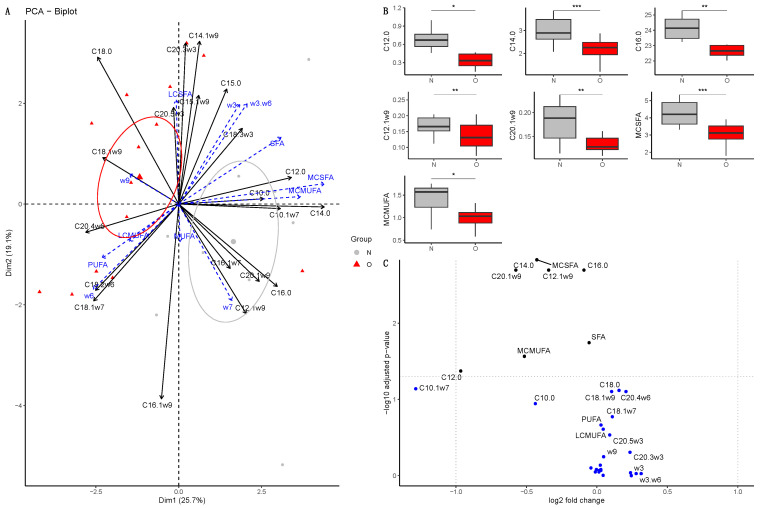
Fatty acid profile in the plasma from foals born to Normal mares (N, n = 14) and Obese mares (O, n = 10) at 30 days of age. (**A**) Biplot of principal component analysis representing the first (40.9%) and second dimensions (15.4%). Variables included in the analysis are represented by the black arrows. Supplementary variables are represented by the blue arrows. Plasma samples from Normal foals are represented by the grey points and ellipse, while plasma samples from Obese foals are represented by the red points and ellipse. The ellipses represented are 95% confidence ellipses around the barycenter. (**B**) Box plot of fatty acids with a significant difference in abundance (% of total fatty acids) between Normal and Obese foals. * = *p* < 0.05, ** = *p* < 0.01, *** = *p* < 0.001. (**C**) Volcano plot representing the log2(fold-change) and the -log10(qvalue). In blue are represented the fatty acids with no significant differences between groups; in black, the fatty acids are significantly different but have a log2(fold-change) between −1 and 1. SFA = saturated fatty acids, MC-SFA = medium-chain saturated fatty acids, LC-SFA = long-chain saturated fatty acids, MUFA = monounsaturated fatty acids, MC-MUFA = medium-chain monounsaturated fatty acids, LC-MUFA = long-chain monounsaturated fatty acids, PUFA = polyunsaturated fatty acids, w3= omega-3 fatty acids, w6= omega-6 fatty acids, w3.w6 = omega-3/omega-6 fatty acids ratio.

**Figure 5 vetsci-10-00691-f005:**
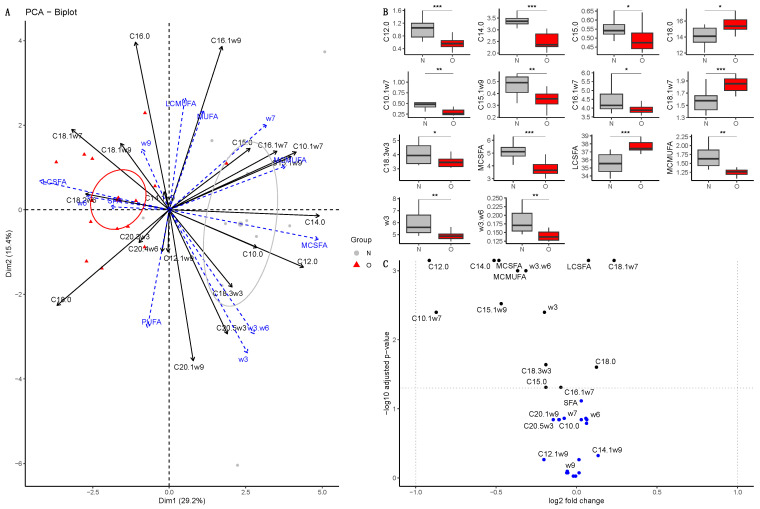
Fatty acid profile in the plasma from foals born to Normal mares (N, n = 14) and Obese mares (O, n = 10) at 90 days of age. (**A**). Biplot of principal component analysis, representing the first (40.9%) and second dimensions (15.4%). Variables included in the analysis are represented by the black arrows. Supplementary variables are represented by the blue arrows. Plasma samples from Normal foals are represented by the grey points and ellipse, while plasma samples from Obese foals are represented by the red points and ellipse. The ellipses represented are 95% confidence ellipses around the barycenter. (**B**). Box plot of fatty acids with a significant difference in abundance (% of total fatty acids) between Normal and Obese foals. * = *p* < 0.05, ** = *p* < 0.01, *** = *p* < 0.001. (**C**). Volcano plot representing the log2(fold-change) and the -log10(pvalue). In blue are the fatty acids with no significant differences between groups; in black, the fatty acids are significantly different but have a log2(fold-change) between −1 and 1. SFA = saturated fatty acids, MC-SFA = medium-chain saturated fatty acids, LC-SFA = long-chain saturated fatty acids, MUFA = monounsaturated fatty acids, MC-MUFA = medium-chain monounsaturated fatty acids, LC-MUFA = long-chain monounsaturated fatty acids, PUFA = polyunsaturated fatty acids, w3= omega-3 fatty acids, w6= omega-6 fatty acids, w3.w6 = omega-3/omega-6 fatty acids ratio.

**Figure 6 vetsci-10-00691-f006:**
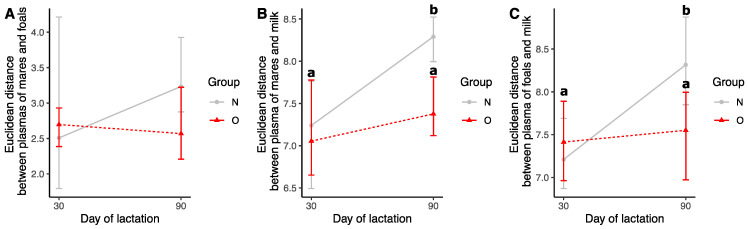
Euclidean distances between the fatty acid profiles of plasma from mares, plasma from foals, and milk at 30 and 90 days of lactation. Different letters indicate a significant difference with a *p* < 0.05. Distances were calculated from individual coordinates in the principal component analysis. (**A**) Distances between plasma from mares and plasma from foals. (**B**) Distances between plasma from mares and milk. (**C**) Distances between plasma from foals and milk.

**Figure 7 vetsci-10-00691-f007:**
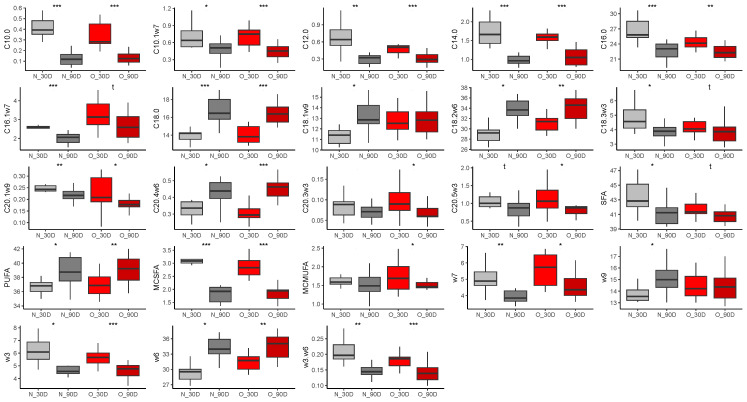
Differences in fatty acid abundance (% of total fatty acids) in plasma from Normal mares (n = 10; grey) and Obese mares (n = 14; red) at 30 days (light colors) and 90 days (dark colors) of lactation. t = *p* < 0.10, * = *p* < 0.05, ** = *p* < 0.01, *** = *p* < 0.001.

**Figure 8 vetsci-10-00691-f008:**
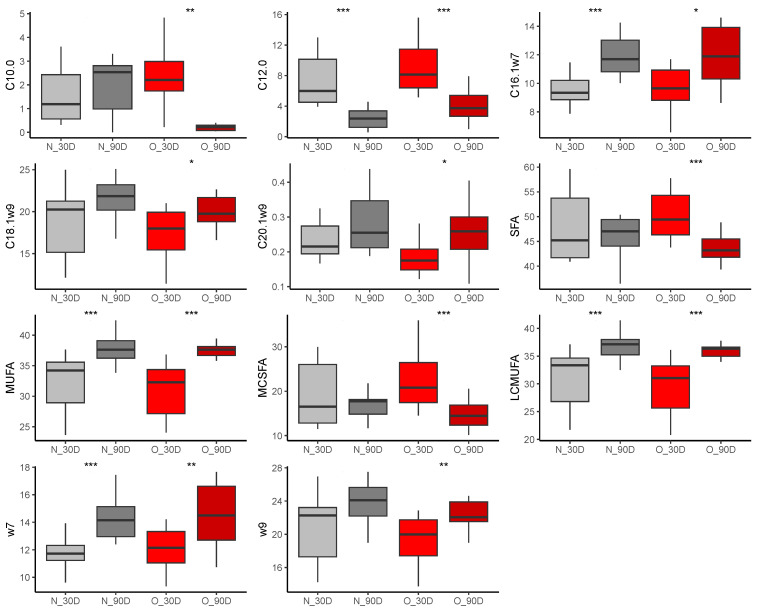
Differences in fatty acid abundance (% of total fatty acids) in milk from Normal mares (n = 10; grey) and Obese mares (n = 14; red) at 30 days (light colors) and 90 days (dark colors) of lactation * = *p* < 0.05; ** = *p* < 0.01; *** = *p* < 0.001.

**Figure 9 vetsci-10-00691-f009:**
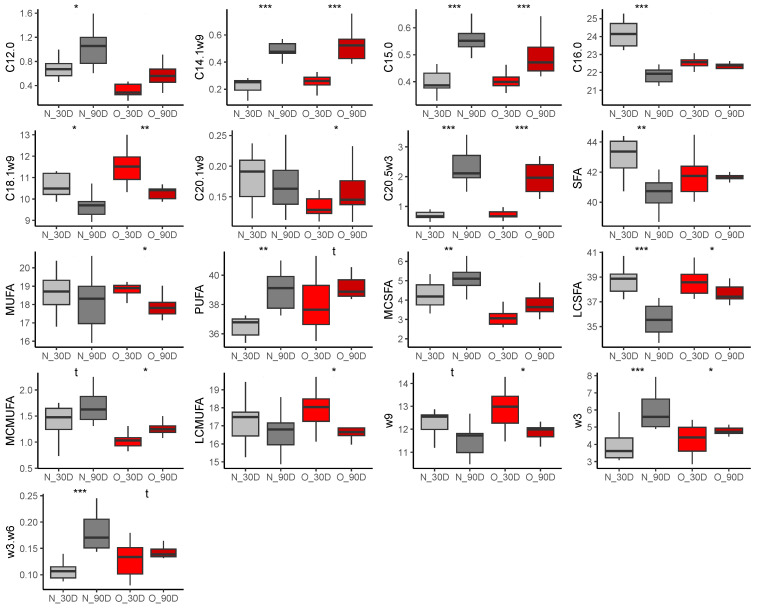
Differences in fatty acid abundance (% of total fatty acids) in plasma from foals born to Normal mares (n = 10; grey) and Obese mares (n = 14; red) at 30 days (light colors) and 90 days (dark colors) of lactation. t = *p* < 0.10, * = *p* < 0.05, ** = *p* < 0.01, *** = *p* < 0.001.

**Figure 10 vetsci-10-00691-f010:**
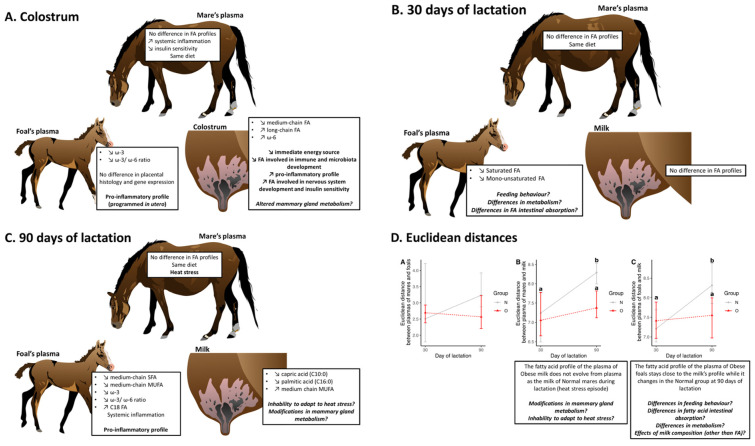
Summary of results observed between plasma of mares, plasma of foals, and colostrum/milk between birth and 90 days of lactation. Different letters indicate a significant difference with a *p* < 0.05.

**Table 1 vetsci-10-00691-t001:** Gene-specific primers and accession number or reference.

	Candidate Gene	Accession Number	Reverse and forward Primers
Reference	*GADPH*	NM_001163856.1	F 5′-AGTTGGGTGCCAAAACTTGTG-3′ R 5′-TGAAGGGGTCATTGATGGCG-3′
	*SCAMP3*	Brosnahan et al. 2012	F 5′-CTGTGCTGGGAATTGTGATG-3′ R 5′-ATTCTTGCTGGGCCTTCTG-3′
	*RPL32*	ENSECAG00000007201	F 5′-TGAAGTGCTGCTCATGTGCA-3′ R 5′-GGGATTGGTGATTCTGATGGC-3′
Vascularization	*ENG*	XM_0011500078.5	F 5′-ACAGTCGAACAGCGACTTCA-3′ R 5′-TTCTTCCCCAAATTCGATTCA-3′
	*Flt1*	XM_003363176.1	F 5′-AGTGTGAGCGGCTCCCTTATG-3′ R 5′-ATGCCAAATGCAGATGCTTG-3′
	*KDR*	XM_001916946.2	F 5′-CAGTGGGCTGATGACCAAGA-3′ R 5′-TCCACCGAAGATTCCATGCC-3′
Nutrient transport	*SCL2A1*	NM_001163971.1	F 5′-TGTGCTCATGACCATCGCC-3′ R 5′-AAGCCAAAGATGGCCACGAT-3′
	*SLC2A3*	XM_001498757.2	F 5′-CCGTTGGTGGTATGATTGGC-3′ R 5′-CAGAACCCCATAAGGCAGCC-3′
	*SlC38A2*	ENSECAT00000016020	F 5′-ACAGCTCGAACAGCGACTTCA-3′ R 5′-TTCTTCCCCAAATTCGATTCA-3′
	*CD36*	ENSECAG00000015229	F 5′-CCGTGCAGAAGCAGTGGTTA-3′ R 5′-CCGTGCAGAAGCAGTGGTTA-3′
	*LPL*	XM_001489577.2	F 5′-AGTTGGGGTGCCAAAACTTGTG-3′ R 5′-GCTTGGTGTACCCCGCAGAC-3′
Growth	*H19*	NR_027326	F 5′-GGACCCCAAGAACCCTCAAG-3′ R 5′-GGGACTTGAAGAAGTCCGGG-3′
	*IGF-2*	NM_001114539	F 5′-TTTCTTGGCTTTTGCCTCGT-3′ R 5′-CCTGCTGAAGTAAAAGCCGC-3′
	*IGF-1R*	XM_001489765.2	F 5′-CGAGAAGACCACCATCAACAAC-3′ R 5′-TGGCAGCACTCGTTTGTTCTC-3′

*GADPH*: Glyceraldehyde 3-phosphate dehydrogenase (reference gene), *SCAMP3*: Secretory Carrier Membrane Protein 3 (reference gene), *RPL32*: Ribosomal Protein *L32* (reference gene), *ENG*: Endoglin (*TGF-β* receptor, involved in angiogenesis), *Flt1*: Fms Related Receptor Tyrosine Kinase 1 (*VEGF* receptor, involved in vasculogenesis and angiogenesis), *KDR*: Kinase Insert Domain Receptor (*VEGF* receptor, involved in vasculogenesis and angiogenesis), *SLC2A1*: solute carrier family 2 member 1 (*GLUT1*, glucose transporter), *SLC2A3*: solute carrier family 2 member 3 (*GLUT3*, glucose transporter), *SLC38A2*: solute carrier family 38 member 2 (*SNAT2*, neutral amino acid transporter), *CD36*: Cluster of differentiation 36 (fatty acid transporter), *LPL*: Lipoprotein Lipase (hydrolyses triglycerides into fatty acids), *H19*: *H19* Imprinted Maternally Expressed Transcript (IncRNA, inhibits growth), *IGF-2*: Insulin-like Growth Factor 2 (growth factor), *IGF-1R*: Insulin-like growth factor 1 receptor (*IGF2* receptor, transduces *IGF2* signal).

**Table 2 vetsci-10-00691-t002:** Common and systematic name, symbol formula and category of fatty acids measured and presented (lipid maps database). Other fatty acids, such as C22:5ω3 and C24:1ω9, were also assayed but not displayed because, for most of the samples, their concentration was below the detection limits.

Common Name	Systematic Name	Symbol Formula	Category
Saturated fatty acid (SFA)			
Capric acid	Decanoic acid	C10:0	Medium-chain fatty acid (MCFA)
Lauric acid	Dodecanoic acid	C12:0	
Myristic acid	Tetradecanoic acid	C14:0	
Pentadecylic acid	Pentadecanoic acid	C15:0	Long-chain fatty acid (LCFA)
Palmitic acid	Hexadecanoic acid	C16:0	
Stearic acid	Octadecanoic acid	C18:0	
Mono-unsaturated fatty acid (MUFA)			
*Medium chain MUFA*			
Caproleic acid	decenoic acid H_2_C = CH(CH_2_)_7_CO_2_H	C10:1	Medium-chain fatty acid (MCFA)
Lauroleic acid	9-dodecenoic acid	C12:1ω3	Medium-chain fatty acid (MCFA)
Myristoleic acid	9-tetradecenoic acid	C14:1ω5	
*Omega-7 fatty acid*			
-Palmitoleic acid	9-Hexadecenoic acid	C16:1ω7	Long-chain fatty acid (LCFA)
Vaccenic acid	11-Octadecenoic acid	C18:1ω7	
*Omega-9 fatty acid*			
6-pentadecenoic acid	6-pentadecenoic acid	C15:1ω9	Long-chain fatty acid (LCFA)
Hypogeic acid	(Z)-hexadec-7-enoic acid	C16:1ω9	
Oleic acid	9Z-octadecenoic acid	C18:1ω9	
Gondoic acid	11Z-eicosenoic acid	C20:1ω9	
Poly-unsaturated fatty acid (PUFA)			
*Omega-3 fatty acid*			
α-Linolenic Acid (ALA)	9Z,12Z,15Z-octadecatrienoic acid	C18:3ω3	Long-chain fatty acid (LCFA)
Eicosatrienoic acid (ETA)	11,14,17-eicosatrienoic acid	C20:3ω3	
Eicosapentaenoic acid (EPA)	5Z,8Z,11Z,14Z,17Z-eicosapentaenoic acid	C20:5ω3	
Docosapentaenoic acid	DPA	C22:5ω3	
*Omega-6 fatty acid*			
Linoleic acid (LA)	9Z,12Z-octadecadienoic acid	C18:2ω6	Long-chain fatty acid (LCFA)
Arachidonic acid (AA)	5Z,8Z,11Z,14Z-eicosatetraenoic acid	C20:4ω6	

## Data Availability

The data is available on the Dryad repository, DOI: https://doi.org/10.5061/dryad.v9s4mw730.

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
