# Peer review of "Obesity during Pregnancy in the Horse: Effect on Term Placental Structure and Gene Expression, as Well as Colostrum and Milk Fatty Acid Concentration"

_vetsci, 2023, doi:10.3390/vetsci10120691_

Round 1

Reviewer 1 Report

Comments and Suggestions for Authors

The paper, titled “Obesity during pregnancy in the horse: effect on term placental 2 structure and gene expression, as well as colostrum and milk 3 fatty acid concentration“ addresses an important and timely topic. I found the subject matter of the article fascinating and read the manuscript with great interest. The paper aligns well with the scope of the journal. However, I believe that in its current form, it has several shortcomings.

This paper investigates the impact of maternal obesity on placental function, milk composition, and foal development in horses. The study is a follow-up to prior research that found maternal obesity during pregnancy led to insulin resistance, systemic inflammation, and osteoarticular lesions in foals. The current study reveals that maternal obesity did not affect placental structure or gene expression. However, it did influence fatty acid profiles in the plasma of foals and mares, colostrum, and milk. Obese mares produced colostrum and milk with a more pro-inflammatory profile, potentially contributing to the developmental programming observed in foals. The study also suggests that maternal obesity could hinder the ability to adapt to heat stress. Overall, this research sheds light on the complex effects of maternal obesity on the next generation of horses, with potential implications for equine health and development.

The paper clearly falls within the scope of the journal as it contributes to our understanding of the impact of maternal obesity on various aspects of equine health and development. The research investigates the effects of maternal obesity on placental function, milk composition, and foal development, which are relevant topics in animal science and veterinary medicine. These findings are valuable for researchers, veterinarians, and equine enthusiasts, aligning well with the journal's focus on advancing knowledge in the field of animal science.

The main question addressed by the research is the impact of maternal obesity on placental function, milk composition, and foal development in horses.

The topic is relevant in the field as it explores the effects of maternal obesity on equine health and development. It addresses a specific gap in the field by focusing on the consequences of maternal obesity in horses, an area with limited previous research.

The research adds to the subject area by providing insights into the effects of maternal obesity on placental development, milk composition, and foal health in horses, which is relatively understudied. This study contributes valuable data and observations.

The abstract correlates with the manuscript content, summarizing the key aspects of the study, its objectives, and the main findings. However, improving the clarity and organization of the abstract could enhance its effectiveness in summarizing the research.

The hypothesis regarding the impact of maternal obesity on equine placental function, milk composition, and foal development is clearly stated. However, it might benefit from more specific predictions or measurable outcomes. This will enhance the testability of the hypothesis and make it easier to draw meaningful conclusions.

The study appears to be well-structured and thorough in many aspects. However, it would be beneficial to provide more detailed information about the specific methods used for data collection, analysis, and statistical procedures. Transparency in these processes is essential for the research's reproducibility and reliability.

To enhance the transparency and replicability of your research, I kindly suggest that you include a section detailing the methods employed for dietary analysis. This should encompass the techniques and procedures used to determine the composition of the diet. Furthermore, I recommend referencing a reputable source for these methods, such as the protocol outlined in 10.3390/ani12141740 and 10.1186/s12917-022-03289-2.

While the study references previous work related to maternal obesity and its effects on foals, it could benefit from a more comprehensive literature review. A deeper exploration of existing research on this topic would help contextualize the findings and provide a clearer understanding of the study's significance within the field.

Consider discussing the economic implications of the research. How might the findings impact horse breeding and healthcare practices? Additionally, explicitly stating the study's limitations is crucial for understanding the scope of its conclusions and recommendations.

The paper briefly mentions that maternal obesity might hinder the ability to adapt to heat stress. Expanding on the practical implications of this finding for horse management, breeding, and healthcare would be valuable for the reader.

Given that there were only ten obese mares in the study, it's important to address the limitations of this small sample size and potential implications for statistical power and generalizability.

It would be beneficial to conclude the paper with a discussion of potential avenues for future research in this area. Identifying unresolved questions or areas that need further exploration can help guide future studies.

The conclusions appear to be consistent with the evidence and arguments presented, and they address the main question posed in the research. However, a more specific and testable hypothesis might strengthen the conclusions.

Author Response

We thank the reviewers for their time and effort to provide insightful comments on our manuscript. We have incorporated the suggestions provided by the reviewers. These suggestions highlighted in the manuscript in blue, and are reported here, along with our responses, in blue. All line numbers refer to the corrected manuscript.

Reviewer 1

The abstract correlates with the manuscript content, summarizing the key aspects of the study, its objectives, and the main findings. However, improving the clarity and organization of the abstract could enhance its effectiveness in summarizing the research.

Some changes were made to the summary to make it clearer as requested. Modifications are highlighted in bold:

“Abstract: In horses, the prevalence of obesity is high and associated with serious metabolic pathologies. Being a broodmare has been identified as a risk factor for obesity. In other species, ma-ternal obesity is known to affect the development of the offspring. This article is a follow-up study of previous work showing that Obese mares (O, n=10, body condition score>4.25 at insemination) were more insulin resistant and presented increased systemic inflammation during pregnancy compared to Normal mares (N, n=14, body condition score<4 at insemination). Foals born to O mares were more insulin resistant, presented increased systemic inflammation and were more affected by osteoarticular lesions. The objective of the present study was to investigate the effect of maternal obesity on placental structure and function, as well as fatty acid profile in plasma of mares and foals, colostrum, and milk until 90 days of lactation, which, to our knowledge, has not been or poorly studied in the horse. Mares of both groups were fed the same diet during pregnancy and lactation. During lactation, mares were housed in pasture. A strong heat wave, followed by a drought, occurred during their 2nd and 3rd months of lactation (summer of 2016 in the Limousin region, France). In the present article, term placental morphometry, structure (stereology) and gene expression (RT-qPCR, genes in-volved in nutrient transport, growth, and development, as well as vascularization) were studied. Plasma of mares and their foals, as well as colostrum and milk were sampled at birth, 30 days, and 90 days of lactation. The fatty acid composition of these samples was measured using gas chromatography. No differences between N and O groups were observed for term placental morphometry, structure, and gene expression. No difference of plasma fatty acid composition was observed between groups in mares. The plasma fatty acid profile of O foals was more pro-inflammatory and indicated an altered placental lipid metabolism between birth and 90 days of age. These results are in line with the increased systemic inflammation and altered glucose metabolism observed until 18 months of age in this group. Colostrum fatty acid profile of O mares was more pro-inflammatory and indicated an increased transfer and/or desaturation of long-chain fatty acids. Moreover, O foals received a colostrum poorer in medium-chain saturated fatty acid, a source of immediate energy for the newborn, that also can play a role in immunity and gut microbiota development. Differences in milk fatty acid composition indicated a decreased ability to adapt to heat stress in O mares, which could have further affected the metabolic development of their foals. In conclusion, maternal obesity affected the fatty acid composition of milk, thus also influencing the foal’s plasma fatty acid composition and likely participating in the developmental programming observed in growing foals.”

The hypothesis regarding the impact of maternal obesity on equine placental function, milk composition, and foal development is clearly stated. However, it might benefit from more specific predictions or measurable outcomes. This will enhance the testability of the hypothesis and make it easier to draw meaningful conclusions. The conclusions appear to be consistent with the evidence and arguments presented, and they address the main question posed in the research. However, a more specific and testable hypothesis might strengthen the conclusions.

L113-118: A more detailed hypothesis has been added: “. Based on the results observed on growing foals and in other species, we hypothesize that placentas from Obese mares will be more inflamed and have an increased expression of glucose transporters compared to Normal placentas. We also hypothesize that the milk of Obese mares will have a more pro-inflammatory fatty acid profile (i.e., in-creased content of omega-6 fatty acids and decreased content of omega-3 fatty acids).”

The study appears to be well-structured and thorough in many aspects. However, it would be beneficial to provide more detailed information about the specific methods used for data collection, analysis, and statistical procedures. Transparency in these processes is essential for the research's reproducibility and reliability.

The authors believe that they have already presented in a satisfactory and detailed manner all the procedures implemented in this study, from the collection of biological samples, their biochemical, histological or molecular analyses, data cleaning and finally the statistical analyses.
Description of bedding during the gestational period has been added line 134: “(straw bedding)”
Moreover, a few details have been added at the request of other reviewers:
L131: “Body condition of the mares was measured on a 0-5 scale”

L143 & 164: “(Vacutainer, BD, USA)”

L171: “using an automat (Varistain, Thermofisher, USA)”

To enhance the transparency and replicability of your research, I kindly suggest that you include a section detailing the methods employed for dietary analysis. This should encompass the techniques and procedures used to determine the composition of the diet. Furthermore, I recommend referencing a reputable source for these methods, such as the protocol outlined in 10.3390/ani12141740 and 10.1186/s12917-022-03289-2.

Details methods and results of dietary analyses have already been provided in our previous article (https://journals.plos.org/plosone/article?id=10.1371/journal.pone.0190309#pone-0190309-g001), in the text, in supplementary data, as well as in the repository Dryad (https://datadryad.org/stash/dataset/doi:10.5061/dryad.s8g04).

To simplify the Material and Methods section, we referred this paragraph to the previous article L 126: “The management of broodmares and foals, as well as dietary analyses, were previously described by Robles et al., 2018[17].”  

While the study references previous work related to maternal obesity and its effects on foals, it could benefit from a more comprehensive literature review. A deeper exploration of existing research on this topic would help contextualize the findings and provide a clearer understanding of the study's significance within the field.

The introduction has been re-written in order to add more information on obesity and maternal effects in other species:

L78-83: “Obesity can also negatively impact sports performances[31], as well as physiological response to exercise and locomotion symmetry[32]. Moreover, an American survey showed that over-conditioned horses costed an average of $434 more per year, for health as well as weight management tools, than non-obese animals [33]. These findings highlight the direct detrimental effects of obesity on horse’s health and well-being, as well as equine economy.”

L 84-89: “In several species, maternal obesity has been shown to affect the health and behaviour of offspring, through mechanisms associated with chronic inflammation in mother and placenta [34–36].For example, offspring from obese mothers have been shown to present an altered glucose metabolism[37] and cardiovascular function[38], an increased fat storage capacity in adipose tissue[39], as well as an increased anxiety and altered food behavior[40,41].  ”

L103-109: “We then demonstrated that maternal obesity affects the horse’s long-term health and development, as observed in other species. Placenta and milk, being both at the interface between the mother and the offspring, might be involved in maternal programming. In the horse, other factors, such as maternal nutrition, age, parity, nursing status or breed have been shown to affect placental structure and function as well as milk composition [4,42,43]. The effects of maternal obesity effects on placental structure and function as well as milk composition are however still un-known in the horse.”

Consider discussing the economic implications of the research. How might the findings impact horse breeding and healthcare practices? Additionally, explicitly stating the study's limitations is crucial for understanding the scope of its conclusions and recommendations. The paper briefly mentions that maternal obesity might hinder the ability to adapt to heat stress. Expanding on the practical implications of this finding for horse management, breeding, and healthcare would be valuable for the reader. Given that there were only ten obese mares in the study, it's important to address the limitations of this small sample size and potential implications for statistical power and generalizability. It would be beneficial to conclude the paper with a discussion of potential avenues for future research in this area. Identifying unresolved questions or areas that need further exploration can help guide future studies.

Two sections have been added in the discussion:

-Study limitations (L851-897):

“This study has several limitations that may have affected the interpretation of the results:

-Sample size: “Ten mares were in the Obese group and 14 in the Normal group. This sample size is considered low and may have led to a low statistical power (especially with the multiple test correction). This could have inflated the risk of false negatives. Therefore, this study needs to be replicated in order to confirm the absence of differences in placental structure and function between obese and non-obese mares, as well as differences in fatty acids in the analyzed tissues between groups.

-Pasture quality information: No information was available for pasture quality, nor pasture consumption by mares and foals during the lactation period. Therefore, the intensity of the drought could not be measured directly on the fields. The fact that, despite having unlimited access to hay, mares of both groups lost body condition during the heat stress episode, highlighted the severity of this event.

-Method of milk sampling: Milk was sampled after a 3-h waiting period, with foals muzzled to prevent them from suckling their mother. Our results may therefore differ from studies for which milk was sampled directly. This waiting period may have decreased the fatty acid concentration and altered the udder fatty acid metabolism. Because all mares and foals were sampled using the exact same protocol, our results are comparable between each other’s, but may not be comparable with other studies.”

-Study perspectives for the equine industry:

“In our initial study, we showed that foals born to obese mares were more susceptible to present osteo-articular lesions and had an altered glucose metabolism [17]. These effects of maternal obesity on the foals’ programming development may therefore be the combination of in utero factors (maternal inflammation and insulin resistance) as well as lactation factors (pro-inflammatory profile of milk from obese mares). The effects of maternal obesity may therefore affect the offspring health and performance at adulthood which could have direct or indirect effects (monetary but also well-being) on the equine industry.

The milk of obese mares shows a decreased nutritional quality compared to normal mares for human consumption. Horse milk can be considered as a substitute of cow-based milk for children with food allergies [109], which would therefore decrease the value of the product.

Moreover, in the present study, obese mares showed a lower ability to adapt to heat stress compared to normal mares. This inability to adapt to heatwave periods can further increase the detrimental effects of maternal obesity on the foals’ development. Effects on maternal health and fertility were not studied in this article and to our knowledge are currently unknown in the horse species. Climate change is intensifying in the world and particularly in Europe. This continent is warming twice as fast as the rest of the world and the occurrence of heatwave and drought events is already increasing above prediction thresholds [110]. These results identify obese mares as more susceptible to heatwaves detrimental effects and provide another argument for a better monitoring of body condition in broodmares.

Finally, to better understand the effects of maternal obesity on the foal’s development, a study of the mothers and foals’ behavior would add both knowledge on the mechanisms involved in programming of the offspring phenotype, as well as potential ways to improve breeding management of obese broodmares.”

Reviewer 2 Report

Comments and Suggestions for Authors

Excellent manuscript significantly adding to the information in the field. Well done. just one tiny query- line 699- is pre pregnant a typo? why would they have colostrum?

lovely summary diagram

Author Response

We thank the reviewers for their time and effort to provide insightful comments on our manuscript. We have incorporated the suggestions provided by the reviewers. These suggestions highlighted in the manuscript in blue, and are reported here, along with our responses, in blue. All line numbers refer to the corrected manuscript.

Just one tiny query- line 699- is pre pregnant a typo? why would they have colostrum?
In the article cited, the authors measured the BMI of women before pregnancy, but not during the pregnancy nor after. L721, we changed the sentence to “In humans, obese women give a colostrum poorer…” to simplify.

Reviewer 3 Report

Comments and Suggestions for Authors

Overall, very interesting research examining the effect of obesity on placenta and milk in mares. A few minor edits are suggested.

Line 67: The word owner is missing (after horse).

Lines 110-111: Please write the BCS score range used in the study

Line 123: “a6ml”, needs spacing.

Line 123: manufacturer of EDTA tubes?

Line 124: “minutecentrifugation”, needs spacing.

Line 149: Manufacturer of hematoxylin/eosin stain?

Line 577-579: The sentence is confusing, please rephrase.

Author Response

We thank the reviewers for their time and effort to provide insightful comments on our manuscript. We have incorporated the suggestions provided by the reviewers. These suggestions highlighted in the manuscript in blue, and are reported here, along with our responses, in blue. All line numbers refer to the corrected manuscript.

Overall, very interesting research examining the effect of obesity on placenta and milk in mares. A few minor edits are suggested.

Line 67: The word owner is missing (after horse).

The word “owners” has been added line 68

Lines 110-111: Please write the BCS score range used in the study

The sentence “Body condition of the mares was measured on a 0-5 scale” line 131.

Line 123: “a6ml”, needs spacing.

Space was added line 143

Line 123: manufacturer of EDTA tubes?

Manufacturer of tubes was added line 143 & 164 “(Vacutainer, BD, USA)”

Line 124: “minutecentrifugation”, needs spacing.

Space was added line 144

Line 149: Manufacturer of hematoxylin/eosin stain?
We do not have this information but we provided the information of the automat used for staining the slides: "using an automat (Varistain, Thermofisher, USA)"  L171

Line 577-579: The sentence is confusing, please rephrase.

Line 599: “Overall, maternal obesity did not affect the measured placental measured: placentas exhibited similarity in size, weight, volume and efficiency with no differences in placental structure nor gene expression.” Has been changed to “Overall, maternal obesity did not affect the placental variables measured: placentas exhibited similarity in size, weight, volume and efficiency with no differences in placental structure nor gene expression.”

Round 2

Reviewer 1 Report

Comments and Suggestions for Authors

The authors have diligently addressed the review comments, significantly enhancing the paper's quality. As a result, it is now well-suited for publication.